# Analysis on BDS-3 Autonomous Navigation Performance Based on the LEO Constellation and Regional Stations

**Baopeng Xu [1], Xing Su [1,\*] , Zhimin Liu [1], Mudan Su [2], Jianhui Cui [1] , Qiang Li [1] , Yan Xu [1], Zeyu Ma [1] and Tao Geng [3]**

1   College of Geodesy and Geomatics, Shandong University of Science and Technology, Qingdao 266590, China; liuzhimin@sdust.edu.cn (Z.L.); jhcui@sdust.edu.cn (J.C.)
2   Beijing Institute of Tracking and Telecommunications Technology, No. 26 Beiqing Road, Beijing 100094, China
3   GNSS Research Center, Wuhan University, Wuhan 430079, China
\*   Correspondence: suxing@sdust.edu.cn; Tel.: +86-532-8605-8006

**Abstract:** The global navigation satellite system (GNSS) is developing rapidly, and the related market applications and scientific research are increasing. Studies based on large low Earth orbit (LEO) satellite constellations have become research hotspots. The global coverage of the LEO constellation can reduce the dependence of navigation satellites on ground-monitoring stations and improve the precise orbit determination (POD) accuracy of navigation satellites. In this paper, we simulate various LEO satellite constellations (with 12, 30, and 60 satellites), along with ground stations' observation data, to examine the impact of LEO satellites on the precision of the BeiDou-3 Global Navigation Satellite System (BDS-3) in terms of its POD accuracy. Using the simulated observation data of both LEO satellites and ground-monitoring stations, we analyze the integrated orbit determination for the LEO and BDS-3 satellites. The findings reveal that the 3D orbital accuracy of BDS-3 is 9.51 dm by using only seven ground-monitoring stations, and it is improved to a centimeter level after adding the LEO constellations. As the number of LEO constellation satellites increases, the impact on improving accuracy gradually diminishes. In terms of time synchronization accuracy in the BDS-3, compared to the results of clock offset using only ground stations, the addition of 12 LEO satellites resulted in an improvement of 49% for RMS1(root mean square) and 52% for RMS2 (standard deviation), the addition of 30 LEO satellites resulted in an improvement of 66% for RMS1 and 70% for RMS2, and the addition of 60 LEO satellites resulted in an improvement of 87% for RMS1 and 90% for RMS2. The integrated orbit determination of the LEO and BDS-3 satellites constellation greatly improves the accuracy of time synchronization. In addition, we also use simulated inter-satellite link (ISL) data to perform enhanced BDS-3 satellites POD and time synchronization experiments. The experiments showed that the orbit determination accuracy of the seven sta (seven stations) and ISL scheme is comparable to that of the seven sta and LEO12 scheme, and that the time synchronization accuracy of the seven sta and ISL scheme is slightly worse. The preliminary experiments showed that the LEO satellite could enhance the orbit determination accuracy of BDS-3 and obtain a higher time synchronization accuracy.

**Keywords:** precise orbit determination; BDS-3; integrated orbit determination; LEO constellations

## 1. Introduction

The precise orbit determination (POD) of navigation satellites is a current research focal point, and it significantly impacts the positioning accuracy of the entire navigation system. BeiDou-3 Global Navigation Satellite System (BDS-3) is a global navigation satellite system (GNSS) developed independently by China [1], which is widely used in transportation, agriculture, forestry and fishery, hydrological monitoring, meteorological measurement, communication timing, power dispatch, disaster relief and mitigation, public security, and other fields [2]. The deployment of the BDS-3 constellation was completed in June

2020. The constellation comprises 3 geostationary orbit (GEO) [3] satellites, 3 inclined geosynchronous orbit (IGSO) satellites, and 24 medium Earth orbit (MEO) satellites. These satellites are capable of autonomous operation, have communication capabilities among themselves, and can operate autonomously without ground station support [4].

POD is a prerequisite for BDS-3 applications, especially in scientific research where high accuracy is required. At present, in the process of orbiting the BDS-3 satellites, the POD of the GEO satellite is a huge challenge [5,6]. Since the geostationary nature of GEO satellites is relative to the ground [7], the observation geometry composed of GEO satellites and ground tracking stations is very poor, and, at the same time, the deployment of ground tracking stations is greatly limited by geography. Compared with medium and high orbit satellites, LEO satellites have the characteristics of low orbital altitude, fast movement speed, and no geographical restrictions on deployment, which can provide a faster change in constellation geometry. The integrated LEO satellites and ground stations complement each other to enhance the accuracy, availability, and integrity of the navigation satellite POD. Geng et al. [8] showed, experimentally, that the geometry for GPS POD is extraordinarily strengthened when the LEO satellites are included as moving tracking stations. This means that we can utilize LEO satellites to improve the accuracy of GPS orbits considerably when there are not enough ground stations. If there are only 21 globally distributed stations plus 3 LEO satellites, the orbits of GPS satellites are still more accurate than when only 43 stations are used. The improvement in GPS orbits is related to the orbit configuration of LEO satellites. The BDS-3 satellites, with inter-satellite link (ISL) equipment, can realize high-precision Ka-band ranging and communication between MEO satellites or between GEO, IGSO, and MEO satellites [9]. The purpose of the ISL is to enable the autonomous orbit determination of navigation satellites and to improve the uneven distribution of ground-monitoring stations for BDS-3 [10]. The use of ISL data and ground station observation data to jointly determine the orbit can improve the orbital accuracy of BDS satellites [4,11,12].

The LEO constellation enhances the orbit accuracy of the BDS-3 constellation, and highly accurate LEO satellite orbits are crucial. Onboard GNSS receivers are now mainly used to determine the orbit of LEO satellites [13], and there are three main types of POD methods for LEO satellites: dynamic orbit determination [14], kinematic precise orbit determination [15], and reduced-dynamic orbit determination [16]. Reduced-dynamic orbit determination adjusts the weight ratio between the satellite's dynamics information and geometric information through process noise. Usually, an additional perturbation force model is introduced during the simplified reduced-dynamic orbit determination to absorb satellite perturbation force model errors and unmodelled errors. Thanks to the International GNSS Service's (IGS) final precise ephemeris and clock products, a post-processing orbiting accuracy of 1–2 cm can be achieved for LEO satellites [17,18].

Integrated orbit determination is the use of ground-based GNSS observations and satellite-based observations to simultaneously solve the GNSS satellite, and the LEO satellite orbit clock-offset, thus improving the overall performance of the navigation satellite [19]. Integrated satellite ground orbit using satellite-based observation data has been performed by scholars. Zhao et al. [20] used PANDA (position and navigation data analysis) software for the integrated determination of GPS and CHAMP satellite orbits, showing that the accuracy of GPS orbit results can reach about 30 mm (using the IGS final precise ephemeris as reference), and the accuracy of CHAMP satellite orbit results is around 114 mm (using the GeoForschungsZentrum Potsdam (GFZ) final precise ephemeris as reference). Duran et al. [21] use GEO/LEO integrated orbit determination, and the improvement in the LEO orbit is obvious. ISL data from multiple GEO satellites and LEO satellites can encrypt continuously trackable arcs, further improving orbit accuracy. Geng et al. [22] used satellite-based GPS observations for integrated orbit determination to significantly improve the GPS satellite orbit accuracy compared to using only ground-based tracking station observations. Lu et al. [4] verified that adding ISL data reduces the dependence on ground stations, and that any improvement in the observation geometry by adding

ISL data has a positive impact on the POD. Haibo Ge et.al. [23] focus on analyzing the LEO enhanced global navigation satellite system (LeGNSS) advantages and challenges for precise orbit and clock determination, precise point positioning (PPP) convergence, Earth rotation parameter estimation, and global ionosphere modeling. Li et al. [24], using LEO satellites with multi-GNSS integrated orbit determination, found that by using 8 globally distributed stations, the accuracy of GNSS satellite orbits can reach the sub-centimeter level, which is comparable to the results of more densely integrated POD, with 65 stations worldwide.

The above work demonstrates that LEO enhances the POD accuracy of GNSS satellites, and that the LEO enhancement method is feasible [25]. However, to the best of the authors' knowledge, there are relatively few studies on the enhancement of the POD accuracy of BDS-3 by large LEO satellite constellations [26,27]. In this paper, constellations containing different numbers of LEO satellites are simulated, and ground-based monitoring station observations, LEO satellite observations, and ISL observations are generated in simulation. Firstly, the effect of ISL data on BDS-3 POD was analyzed by using the ISL and ground-station-integrated orbit determination, and secondly, the influence of different LEO satellite constellations on enhancing BDS-3 satellites POD was discussed by using LEO satellite and ground-station-integrated orbit determination. The second section details the processing method and orbit determination strategy of integrated orbit determination between the simulated LEO constellation and the ground station. Then, in the third section, the experiment of integrated orbit determination between the ISL and the ground station, and the experiment of integrated orbit between the LEO satellites and ground station, were carried out, and the POD accuracy and time synchronization accuracy of BDS-3 satellites were analyzed. The results are discussed in Section 4. Conclusions are presented in Section 5.

## 2. BDS-3 Satellites POD Method

Satellite orbit determination techniques based on ground-based monitoring station observations are now the most important way to obtain precise orbits of navigation satellites. Unlike ground-based monitoring stations, the main purpose of loading GNSS receivers on LEO satellites is for LEO satellite orbit determination. When the precise orbit of the GNSS satellite is used as a known quantity, the orbit of the LEO satellite is obtained by using the satellite-based GNSS observation data. When the orbit of the GNSS satellite is unknown, the LEO satellite can be used as a space-based observation-data-receiving station, and their data can be jointly used with ground monitoring station data for precise GNSS satellite orbit calculation.

### 2.1. Satellite Equations of Motion and Observation Models

Following Newton's theorem of motion, satellites are subjected to a combination of multiple perturbing forces. Therefore, the equations of motion need to be established in an inertial coordinate system. Combining the various perturbing forces, the equations of motion of a navigation satellite can be written as [28]:

$$\ddot{r}(t) = -\frac{GM_E}{r^3}r + f_1(r, \dot{r}, p, t) = f_0(r, t) + f_1(r, \dot{r}, p, t) \tag{1}$$

where $t$ is for a specific moment of observation, $r$, $\dot{r}$, $\ddot{r}$ are the position, velocity, and acceleration of the satellite's center of mass at the moment $t$, respectively. $p$ is the vector of model parameters to be estimated with the POD filter. $f_0$ represents the gravitational force of the Earth. $f_1$ indicate other perturbations (spherical harmonic expansion of the Earth's gravity field, non-gravitational perturbations, third-body perturbations, etc).

The second-order differential equation can be written as a system of first-order differential equations. The form is as follows:

$$\begin{cases} \dot{r} = v \\ \dot{v} = \ddot{r} \\ \dot{p} = 0 \end{cases} \tag{2}$$

The initial state of the satellite can be recorded as:

$$\begin{cases} r(t_0) = v \\ v(t_0) = \dot{r} \\ p(t_0) = p_0 \end{cases} \tag{3}$$

The first-order differential equation can be written as:

$$\begin{cases} \dot{X} = F(X,t) \\ X(t_0) = X_0 \end{cases} \tag{4}$$

By solving the satellite variational equation, the state transfer matrix can be derived so that the satellite's precise orbital parameters can be obtained from the initial orbital parameters. Assuming that $X^*$ is the satellite reference orbit and linearizing the differential Equation (4) (neglect the higher order terms), we obtain:

$$\dot{X} = F(X^*,t) + \left. \frac{\partial F(X,t)}{\partial X} \right|_* (X - X^*) = \dot{X}^* + \left. \frac{\partial F(X,t)}{\partial X} \right|_* (X - X^*) \tag{5}$$

of which,

$$\frac{\partial F(X,t)}{\partial X} = \begin{bmatrix} \dfrac{\partial \dot{r}}{\partial r} & \dfrac{\partial \dot{r}}{\partial \dot{r}} & \dfrac{\partial \dot{r}}{\partial p} \\ \dfrac{\partial \ddot{r}}{\partial r} & \dfrac{\partial \ddot{r}}{\partial \dot{r}} & \dfrac{\partial \ddot{r}}{\partial p} \\ \dfrac{\partial \dot{p}}{\partial r} & \dfrac{\partial \dot{p}}{\partial \dot{r}} & \dfrac{\partial \dot{r}}{\partial p} \end{bmatrix} = \begin{bmatrix} 0 & I & 0 \\ \dfrac{\partial \ddot{r}}{\partial r} & \dfrac{\partial \ddot{r}}{\partial \dot{r}} & \dfrac{\partial \ddot{r}}{\partial p} \\ 0 & 0 & 0 \end{bmatrix} \tag{6}$$

$x(t) = X - X^*, A(t) = \dfrac{\partial F(X,t)}{\partial X}$, Equation (5) can be expressed as:

$$\dot{x}(t) = A(t) \cdot x(t) \tag{7}$$

This can be recorded as:

$$x(t) = \psi(t, t_0) \cdot x_0 \tag{8}$$

$$\psi(t, t_0) = \begin{bmatrix} \dfrac{\partial r}{\partial r_0} & \dfrac{\partial r}{\partial \dot{r}_0} & \dfrac{\partial r}{\partial p} \\ \dfrac{\partial \dot{r}}{\partial r_0} & \dfrac{\partial \dot{r}}{\partial \dot{r}_0} & \dfrac{\partial \dot{r}}{\partial p} \\ \dfrac{\partial p}{\partial r_0} & \dfrac{\partial p}{\partial \dot{r}_0} & \dfrac{\partial p}{\partial p} \end{bmatrix} \tag{9}$$

Further decomposition of $\psi(t, t_0)$:

$$\phi(t, t_0) = \frac{\partial x}{\partial x_0} = \begin{bmatrix} \dfrac{\partial r}{\partial r_0} & \dfrac{\partial r}{\partial \dot{r}_0} \\ \dfrac{\partial \dot{r}}{\partial r_0} & \dfrac{\partial \dot{r}}{\partial \dot{r}_0} \end{bmatrix}_{6 \times 6} \tag{10}$$

$$S(t) = \frac{\partial x}{\partial p} = \begin{bmatrix} \dfrac{\partial r}{\partial p} \\ \dfrac{\partial \dot{r}}{\partial p} \end{bmatrix} \tag{11}$$

Additionally, there are:

$$\begin{aligned} \phi(t_0, t_0) &= I_{6*6} \\ S(t_0) &= 0_{6*n_d} \end{aligned} \tag{12}$$

where $n_d$ denotes the vector dimension corresponding to the relevant perturbing force and $I_{6*6}$ denotes the unit matrix. $\phi(t, t_0)$ is the state transfer matrix, which can represent the partial derivatives of the position and velocity of the satellite at any moment concerning the initial partial derivative. $S(t)$ is a sensitivity matrix, that describes the partial derivatives of the state vector and dynamical parameters. In the actual solution process, the state transfer matrix $\phi(t, t_0)$ is usually obtained by the numerical integration method due to the complexity of the relevant perturbing force function on the satellite.

The GNSS raw observation equations are expressed in terms of pseudorange and carrier phase, depending on the amount of observation:

$$P_{i,r}^s = \rho_r^s + c\delta t_r - c\delta t^s + \mu_i I_r^s + Rm_{P_i,r}^s + \varepsilon_{p_i} \tag{13}$$

$$L_{i,r}^s = \rho_r^s + +c\delta t_r - c\delta t^s - \mu_i I_r^s + \lambda_i N_{i,r}^s + Rm_{P_i,r}^s + \varepsilon_{L_i} \tag{14}$$

where: $s$ indicates satellite, $r$ indicates Earth station, $\rho_r^s$ is the geometric distance of the station from the satellite, $i$ is a pseudorange or a frequency point of the carrier phase, and $\delta t$ is the clock offset; $\mu_i$ is the ionospheric delay factor at the $i$ frequency point and $I_r^s$ is the ionospheric delay corresponding to the observed quantity; $\lambda_i$ indicates the wavelength of the frequency $i$, $N_{i,r}^s$ is the ambiguity parameter, $Rm$ are tropospheric delays, relativistic effects, multipath effects, antenna phasecenter errors, and other errors; and $\varepsilon_{p_i}$ and $\varepsilon_{L_i}$ are the pseudorange and phase observation noise, respectively.

Usually, instead of using the original observations directly, various combinations of pseudorange and carrier phase are used to eliminate or attenuate the associated errors. Combinations of measurements that eliminate the effect of the first-order term in the ionosphere are the most widely used. In this case, the combined pseudorange and carrier phase quantities can be expressed as:

$$\begin{aligned} P_{LC,r}^S &= \frac{f_1^2}{f_1^2 - f_2^2} P_{1,r}^S - \frac{f_2^2}{f_1^2 - f_2^2} P_{2,r}^S \\ L_{LC,r}^S &= \frac{f_1^2}{f_1^2 - f_2^2} L_{1,r}^S - \frac{f_2^2}{f_1^2 - f_2^2} L_{2,r}^S \end{aligned} \tag{15}$$

where $P_i$, $L_i$ ($i = 1, 2$) indicates pseudorange and carrier phase observations and $f_i$ represents different frequencies.

Assume that the coordinates of the station under the Earth-fixed coordinate system are $X_{r,*} = (x_{r,*}, y_{r,*}, z_{r,*})$. The first-order Taylor series expansion of the satellite on $X^{s,*} = (x^{s,*}, y^{s,*}, z^{s,*})$ reference orbit. $X^{s,*}$ is defined as the reference orbit of the satellite [17,18,29]. According to Equations (1) and (14), the linearized equation can be obtained as:

$$\begin{bmatrix} v_{PC} \\ v_{LC} \end{bmatrix} = \begin{bmatrix} -u_r^s & u_r^s & 1 & 1 & -1 & 0 \\ -u_r^s & u_r^s & 1 & 1 & -1 & 1 \end{bmatrix} X_{all} - \begin{bmatrix} l_{PC} \\ l_{LC} \end{bmatrix} \tag{16}$$

where $v_{PC}$ is the pseudorange observation error, $v_{LC}$ is the carrier phase observation error, and $l_{PC}$ and $l_{LC}$ are the difference between the estimate of the pseudorange and the carrier phase observation. The coordinates of the ground-monitoring station in the inertial system can be expressed as $X'_{r,*} = R \cdot X_{r,*}$. $R$ denotes the corresponding trans-

formation matrix of the coordinate system, $u_r^s = \begin{bmatrix} \dfrac{x'_{r,*} - x^{s,*}}{\rho_{r,*}^s} & \dfrac{y'_{r,*} - y^{s,*}}{\rho_{r,*}^s} & \dfrac{z'_{r,*} - z^{s,*}}{\rho_{r,*}^s} \end{bmatrix}$, $X_{all} = \begin{bmatrix} dX^s & dX_r & dRm & t_r & t^s & N \end{bmatrix}^T$, and $X_{all}$ is the transpose matrix of all the correction values. $dX^s$ is the correction of the satellite position, velocity, and perturbing force model, etc., concerning the reference orbit.

For the POD of navigation satellites, the equations of motion and the equations of observation correspond to the dynamical and geometrical observation information of the satellite, respectively, and the solution of the orbit is to use these two aspects of information to solve for the parameters related to the initial state of the satellite.

Suppose that, at the moment $t_i$ the observation noise is $V_i = \begin{bmatrix} v_{PC}(t_i) \\ v_{LC}(t_i) \end{bmatrix}$, the difference between the observed quantity and its valuation is $L_i = \begin{bmatrix} l_{PC}(t_i) \\ l_{LC}(t_i) \end{bmatrix}$; the vector of orbital parameters to be estimated is $x(t_i) = dX^s$. Make $H_i = \begin{bmatrix} -u_r^s \\ -u_r^s \end{bmatrix}$, the other parameter vectors to be estimated are $y_i = \begin{bmatrix} dX_r & dRm & t_r & t^s & N \end{bmatrix}^T$, and make $B_i = \begin{bmatrix} u_r^s(t_i) & 1 & 1 & -1 & 0 \\ u_r^s(t_i) & 1 & 1 & -1 & 0 \end{bmatrix}$, where $H_i$ and $B_i$ are the design matrices for the parameter vector $x(t_i)$ and $y_i$, respectively. Substituting into Equation (17) gives:

$$v_i = H_i x(t_i) + B_i y_i - l_i \tag{17}$$

Combining Equations (8) and (17) yields:

$$v_i = H_i x(t_i) + B_i y_i - l_i = H_i \psi(t_i, t_0) x_0 + B_i y_i - l_i \tag{18}$$

Make $A_i = \begin{bmatrix} H_i \psi(t_i, t_0) & B_i \end{bmatrix}$; $x_i = \begin{bmatrix} x_0 \\ y_i \end{bmatrix}$. Substituting into Equation (18) gives:

$$v_i = A_i x_i - l_i \tag{19}$$

The observations of all satellites from $t_1$ to $t_n$ within the observation arc can be expressed in Equation (19), after which a least squares processing algorithm is used to perform repeated iterations of the solution until the result of the solution reaches the set limits.

Using the orbits and clock offset of LEO and BDS satellites as unknown parameters, the use of LEO satellite on-board data can effectively increase the number of observable arc segments and enhance the geometrical structure of station satellites, thus improving the orbit determination accuracy of BDS satellites.

$$L_{sta} = G(x_{bds}, x_{sta}, x_0, t) + v_{sta} \tag{20}$$

$$L_{leo} = G(x_{bds}, x_{sta}, x_0, t) + v_{leo} \tag{21}$$

where $L_{sta}$ and $L_{leo}$ denotes BDS observations obtained by ground-based and satellite-based receivers, respectively, $x_{bds}$ and $x_{leo}$ denote the BDS orbital parameters and LEO satellite orbital parameters, respectively, $x_{sta}$ denotes the station-related parameter, $x_0$ denotes the parameters related to the observed quantity, such as the carrier phase ambiguity, clock offset parameters, etc., and $v_{sta}$ and $v_{leo}$ correspond to the station and LEO satellite measurement noise, respectively.

The observation equation is linearized and written in matrix form, denoted as:

$$l = \begin{bmatrix} l_{sta} & l_{leo} \end{bmatrix}^T \tag{22}$$

$$A = \begin{bmatrix} \dfrac{\partial G}{\partial x_{bds}} & 0 & \dfrac{\partial G}{\partial x_{sta}} & \dfrac{\partial G}{\partial x_0} \\ \dfrac{\partial F}{\partial x_{bds}} & \dfrac{\partial F}{\partial x_{leo}} & 0 & \dfrac{\partial F}{\partial x_0} \end{bmatrix} \tag{23}$$

$$\delta x = \begin{bmatrix} \delta x_{bds} & \delta x_{leo} & \delta x_{sta} & \delta x_0 \end{bmatrix}^T \tag{24}$$

The corresponding least squares solution can be expressed as:

$$\delta \hat{x} = \left(A^T P A\right)^{-1} A^T P l \tag{25}$$

$$\hat{Q}_x = \left(A^T P A\right)^{-1} = \left(A_{sta}^T P_{sta} A_{sta} + A_{leo}^T P_{leo} A_{leo}\right)^{-1} \tag{26}$$

where $P = \begin{bmatrix} P_{\text{sta}} & \\ & P_{leo} \end{bmatrix}$, $P$, $P_{\text{sta}}$, and $P_{leo}$ represent the weight matrix, ground-based weight matrix, and satellite-based receivers weight matrix. $\hat{Q}_x$ is a covariance matrix.

### 2.2. Processing Strategy

The satellite orbit determination mainly includes the choice of observation model parameters, the orbit error correction model, and the satellite dynamic model. The important options of the detailed processing strategy for the LEO POD are listed in Table 1.

Due to the large number of observations to be processed, we selected an arc length of 72 h and a processing interval of 300 s. The elevation angle threshold cut-off elevation is set to 5° and 1° for ground stations and LEO satellites, respectively [30]. In terms of force model, BDS-3 and LEO satellites suffer from different perturbative forces, since they move at different orbital altitudes, especially in the aspect of non-gravitational forces. For BDS-3 satellites, the solar radiation pressure (SRP) serves as the primary source of non-gravitational forces, and the atmospheric drag is neglected in the BDS-3 POD processing [31]. Different from BDS-3 satellites, the atmosphere drag plays a dominant role in the non-gravitational forces for LEO. We used the DTM94 model for this purpose. The estimated model parameters in a POD fit analysis include the satellite state vector, clock offsets, empirical accelerations, SRP, and atmospheric drag. The estimation process involves a least squares adjustment method, whereby the objective is to minimize the difference between the observed and modeled data by iteratively adjusting the model parameters until a satisfactory fit is achieved [32].

**Table 1.** Detailed processing strategy for the LEO POD.

| Project | Parameters and Models |
|---|---|
| Elevation Angle Threshold | 5° for the ground station and 1° for LEO |
| Earth Gravity Field | EIGEN6C (12 × 12) for BDS-3 and EIGEN6C (120 × 120) for LEO [33] |
| N-body Perturbation | JPL DE405 [34] |
| SRP | ECOM 5 model for BDS-3 and macro-model for LEO [24] |
| Atmospheric drag | DTM94 [35] for LEO |
| BDS-3 Phase Center Offset (PCO) and Phase Center Variation (PCV) | igs14.atx |
| Station PCO and PCV | igs14.atx |
| LEO PCO and PCV | None |
| Solid tide and Pole tide | IERS 2010 [36] |
| Relativity | IERS 2010 |
| Ocean tide | FES 2004 [37] |
| Earth rotation parameters | One set per arc |

The whole constellation rotation problem is solved using a ground-based anchoring station, which is located in China and can be linked to all BDS-3 satellites for dual one-

way ranging, and its coordinates are fixed to provide the constellation reference. Since the anchor station and the satellite share the same ISL measurement and communication system, the Ka observations of the anchor station and the inter-satellite Ka observations are treated in the same way. The important options of the detailed processing strategy for the ISL POD are listed in Table 2.

**Table 2.** Detailed processing strategy for the ISL POD.

| Project | Parameters and Models |
|---|---|
| Earth Gravity Field | EIGEN6C (12 × 12) for BDS-3 |
| N-body Perturbation | JPL DE405 |
| SRP | ECOM 5 model for BDS-3 |
| BDS-3 Phase Center Offset (PCO) and | igs14.atx |
| Station PCO and PCV | igs14.atx |
| Solid tide and Pole tide | IERS 2010 |
| Relativity | IERS 2010 |
| Ocean tide | FES 2004 |
| Earth rotation parameters | One set per arc |
| Orbit parameters to be estimated | Satellite initial position, velocity and solar pressure parameters |

## 3. Results

In this section, the accuracies of BDS-3 POD and time synchronization are analyzed using ground station data and ISL data. Joint orbit determination experiments with different LEO satellite constellations and ground stations were also carried out. The factors affecting the orbital accuracy and time synchronization of the BDS-3 satellites are discussed in detail.

### 3.1. BDS3 Joint POD Results Based on Ground Monitoring Stations and ISL

In this section, we explore the use of ISL data and ground station data to validate the POD and time synchronization capabilities of the BDS-3 constellation. BDS-3 employs ISL technology, whereby each satellite carries ISL equipment that allows two-way communication and measurements in Ka-band between satellites, or between satellites and ground stations (anchor stations) equipped with the same equipment [38]. Broadcast ephemerides can be updated using ISL technology, allowing the autonomous navigation of the BDS-3 constellation to be completely independent of ground-monitoring stations. Ka-band pseudorange measurements between the BDS-3 satellites, and between the satellites and the anchor station, provide the independent orbit and time synchronization capabilities of the ground monitoring stations. We divided the complete POD simulation process into four parts. Firstly, we converted the Kepler orbit parameter to the position and velocity in J2000 inertial system (namely the initial conditions) at the beginning epoch. Secondly, the initial conditions were extended to three days by orbit integration, in this process, we set the force model to the parameters in Table 2. Next, BDS's ground monitoring stations and ISL observation data were simulated based on the satellite orbit inherited by step two. Finally, in the POD process, the estimated parameters fitted by POD included BDS-3 satellites position and velocity, solar radiation pressure parameters, satellite clock bias, zenith tropospheric delay (for anchor stations), and receiver clock bias.

Firstly, we performed a joint POD experiment using the seven sta (seven stations) and seven sta and ISL schemes with an orbital arc of 3 days. The seven sta scheme uses only ground stations and no ISL data, while the seven sta and ISL scheme uses ground stations and BDS-3 ISL data for a joint POD to analyze BDS-3 POD and time synchronization.

The observation data of the BDS-3 ground station are simulated, the elevation angle threshold is set to 5 degrees. The pseudorange measurement is added with a random noise of 1.000 m and a systematic error of 0.030 m, while the carrier-phase measurement is added with random noise of 0.002 m and a systematic error of 0.030 m [29,39]. We simulate BDS-3 ISL observations between GEO and MEO, IGSO and MEO, MEO and MEO, and IGSO and IGSO. The satellite geometry visibility condition is judged as the ISL height above 1000 km

on the Earth's surface. We added 0.100 m random noise and 0.100 m systematic error to the ISL observation values [11].

We simulated seven ground-monitoring stations, located in Harbin, Beijing, Xi'an, Urumqi, Lhasa, Shanghai, and Sanya, as shown in Figure 1.

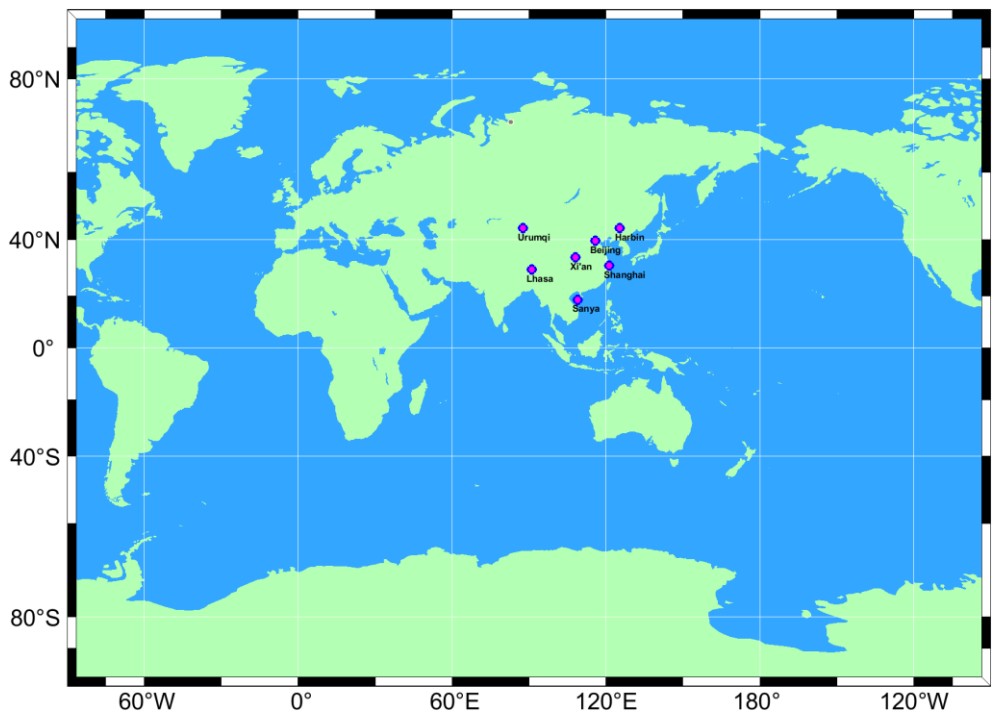

**Figure 1.** Distribution of the 7 ground monitoring stations.

The BDS-3 satellites adopt the 3 GEO + 3 IGSO + 24 MEO constellation configuration. The three GEO satellites are positioned over the equator at 80°E, 110.5°E, and 140°E, respectively. The three IGSO satellites have a figure-of-eight ground-track, with their intersection point over the equator at 120°E, and their orbital planes differ by 120°E from each other in space. The 24 MEO satellites adopt the Walker: 24/3/1 constellation configuration, have an orbital radius of 27,906 km, and are evenly distributed in 3 orbital planes with 120 degrees of difference in equatorial longitude from the ascending node [1], as shown in Table 3. Figure 2 shows the BDS-3 satellites' ground-tracks.

**Table 3.** Nominal constellation configuration of BDS-3.

| GNSS | BDS-3 Satellites | | |
|---|---|---|---|
| Orbit Type | GEO | IGSO | MEO |
| Satellites number | 3 | 3 | 24 |
| Pseudo-random noise (PRN) number | C01, C02, C03 | C04, C05, C06 | C07–C30 |
| Altitude | 35,786 km | 35,786 km | 21,528 km |
| Inclination | 0° | 55° | 55° |
| Constellation | Located at 80°E, 110.5°E, and 140°E | RAAN of 118°E | Walker (24/3/1) |

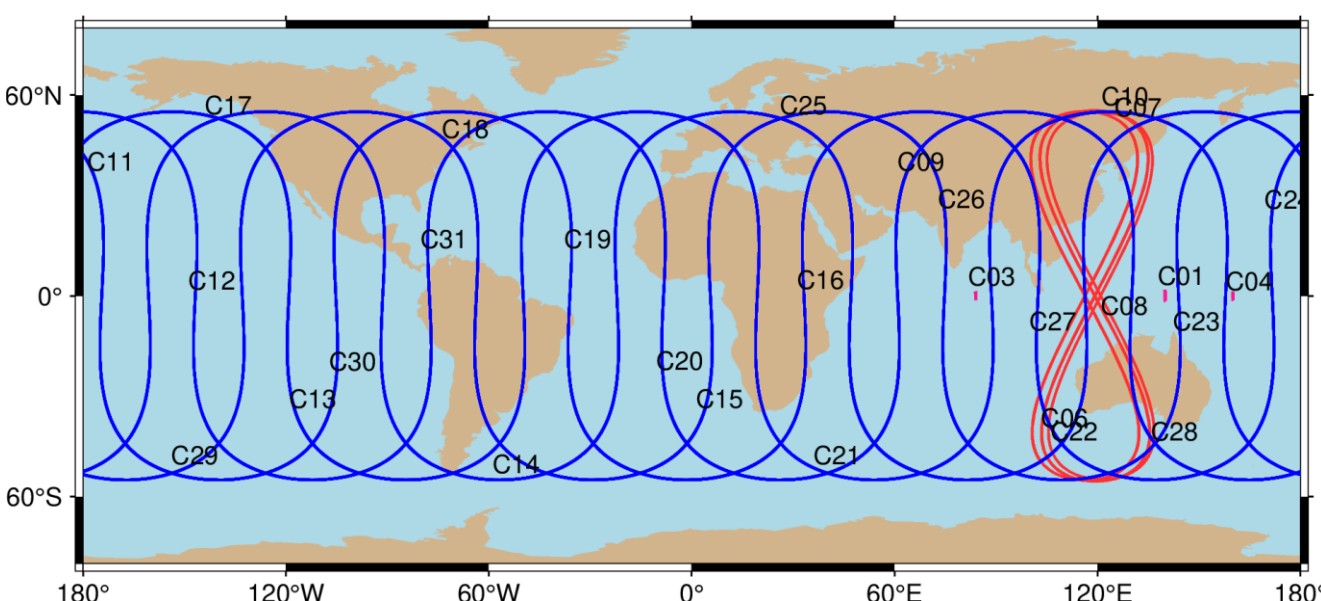

**Figure 2.** BDS-3 constellation navigation subastral point ground-tracks diagram.

A satellite orbit assessment uses the difference between the calculated navigation satellite orbit and the real orbit to count the RMS value of the orbit difference in the along-track, cross-track, radial (A, C, R), and three-dimensional (3D) directions of the BDS-3 satellites' orbit [10].

Figure 3 shows the effective values of seven sta and seven sta and ISL in the 3D RMS, using both schemes for POD and time synchronization experiments. It can be seen that the average orbit accuracy of all BDS-3 satellites is 0.748 m without ISL data. The seven sta and ISL scheme has an average orbital accuracy of 0.035 m. The use of ISL data resulted in a high orbit accuracy for BDS-3, with an improvement of about 96.1% compared to the seven sta schemes. The addition of ISL data can make up for the lack of the number and distribution of ground-monitoring stations, improve the tracking of geometric observation shape changes, and make the satellite geometry more robust [40].

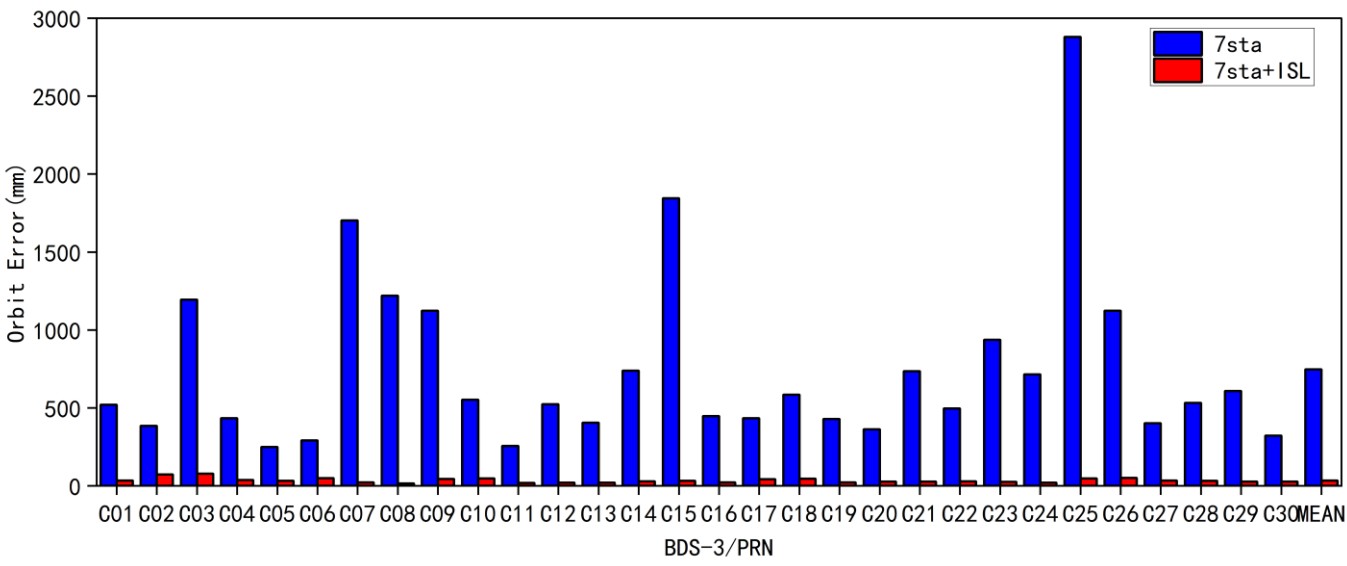

**Figure 3.** RMS in 3D direction for both 7 sta and 7 sta and ISL schemes.

Figures 4 and 5 show the RMS in the A, C, and R direction for the seven sta and seven sta and ISL schemes, respectively. The average RMS of all BDS-3 satellites in the seven

sta scheme were 0.679 m, 0.231 m, and 0.152 m in the A, C, and R directions, respectively, while the average RMS for all BDS-3 satellites in the seven sta and ISL scheme is 0.025 m in the along direction, 0.022 m in the cross direction, and 0.005 m in the radial direction. The seven sta and ISL scheme improves 96.3%, 90.5%, and 96.8% in the A, C, and R directions compared to the seven sta scheme [40].

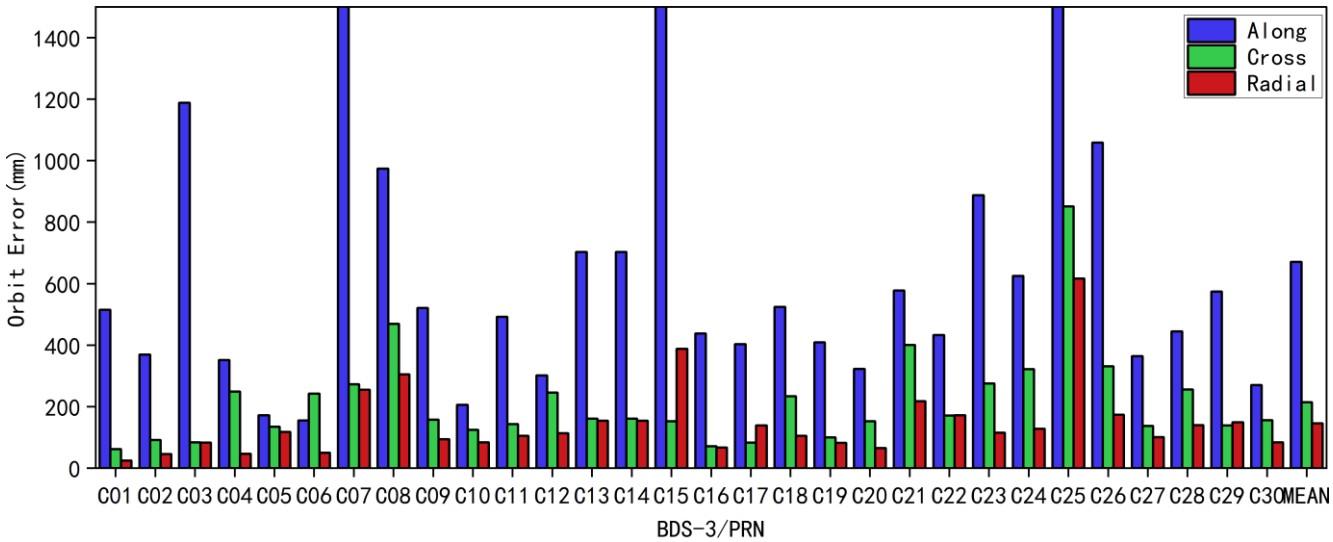

**Figure 4.** The 7 sta scheme BDS-3 constellation in A, C, and R directions of RMS.

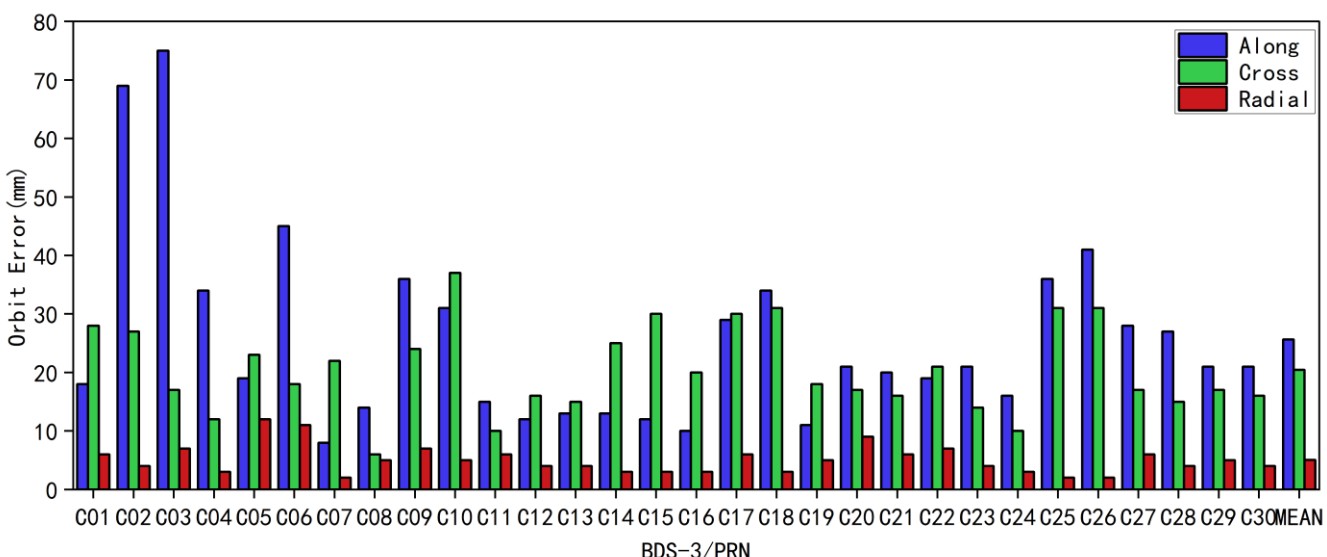

**Figure 5.** The 7 sta and ISL scheme BDS-3 constellation in A, C, and R directions of RMS.

The satellite clock offset accuracy assessment method uses the second difference comparison method [41,42]. To begin with, the C01 satellite is designated as the reference clock. The clock offset outcomes obtained from other satellites are then referenced to this clock, thereby eliminating any differences in clock offset caused by contrasting reference clocks. Finally, a double difference comparison is conducted between the obtained results. The RMS1 and RMS2 of the statistical quadratic difference time series are given by Equation (27):

$$RMS1 = \sqrt{\left(\sum_{i=1}^{n} \Delta_i \Delta_i\right)/n}, \ RMS2 = \sqrt{\left(\sum_{i=1}^{n} (\Delta_i - \overline{\Delta})(\Delta_i - \overline{\Delta})\right)/n} \tag{27}$$

where $\Delta_i$ is the quadratic difference at the epochs and $\overline{\Delta}$ is its average value. RMS1 and RMS2 are the roots mean square and standard deviation of the second difference time series [43].

As shown in Figures 6 and 7, using the seven sta scheme, the average accuracy of RMS1 for all satellites of BDS-3 is 0.89 ns, and the average accuracy of RMS2 is 0.83 ns, respectively. In the case of only seven ground stations, the time synchronization accuracy is on average 2.9 times worse than the seven sta and ISL scheme, which is mainly related to the distribution and number of ground stations. We can see from Figure 2 that the ground trajectories of MEO satellites are distributed globally, and that ground monitoring-stations are concentrated in China. Therefore, when the satellites are abroad, the broadcast ephemerides of the ground stations cannot be injected and updated into MEO satellites, resulting in the poor time synchronization accuracy of MEO. Therefore, it is normal for the MEO satellites in Figure 6 to have a lower time synchronization accuracy than the IGSO and GEO satellites.

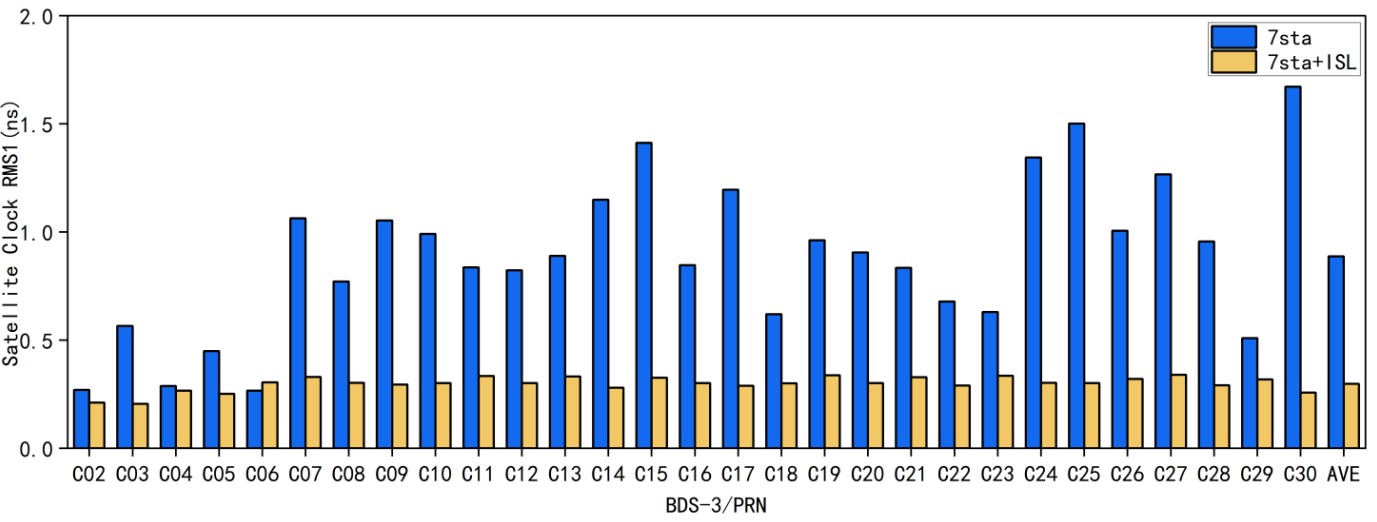

**Figure 6.** RMS1 for the 7 sta and 7 sta and ISL scheme BDS-3 time synchronization case.

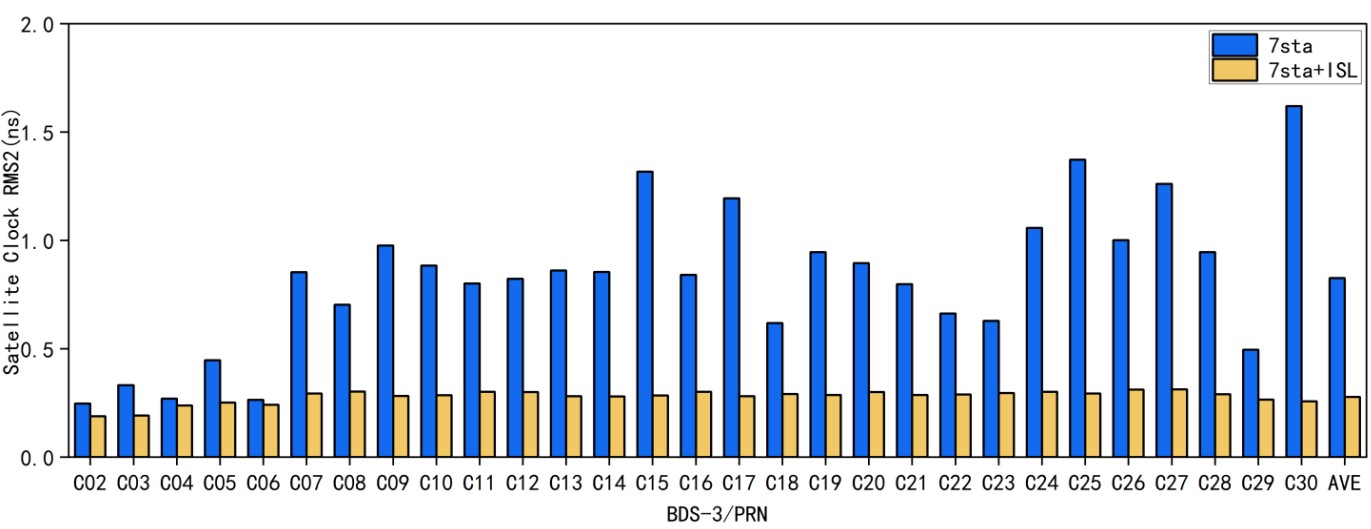

**Figure 7.** RMS2 for the 7 sta and 7 sta and ISL scheme BDS-3 time synchronization case.

As shown in Figures 6 and 7, the average RMS1 and RMS2 of all BDS-3 satellites are 0.3 ns and 0.29 ns for the time-synchronized experiment on the seven sta and ISL scheme, respectively. Compared with the seven sta scheme, the use of ISL can improve the time synchronization accuracy of BDS-3 satellites.

### 3.2. BDS-3 Joint POD Results Based on Ground-Monitoring Stations and LEO Satellites

To analyze the effect of different numbers of LEO satellites on BDS-3's space signal [44] enhancement, we simulated three types of LEO satellite constellations, namely LEO12, LEO30, and LEO60. The orbital altitude of all LEO satellites is 975 KM, and they are equally distributed in six orbital planes namely Walker 12/6/1, Walker 30/6/1, and Walker 60/6/1, with an orbital inclination of 55°.

Both analog ground stations and LEO satellite-monitoring stations can receive navigation signals from the BDS-3 satellites navigation constellation to obtain carrier and pseudorange observation of all visible satellites in the epoch. We added 1.000 m of random noise and 0.030 m of pseudorange systematic error [29,39], as well as 0.002 m of random noise and 0.030 m of carrier system error, to the simulated observations data, and the orbital arc is 3 days. As shown in Figure 8, the average 3D accuracies of all LEO satellite orbits in 3D are 1.7 cm, 1.6 cm, and 0.6 cm, respectively.

We used four different schemes, of 7 sta, 7 sta and 12 LEO, 7 sta and 30 LEO, and 7 sta and 60 LEO, for the distribution of the number of LEO satellites for the POD, and counted the average 3D accuracies. The average 3D accuracies of the four schemes were 95.1 cm, 3.1 cm, 2.4 cm, and 2.0 cm, respectively, as shown in Figures 9 and 10.

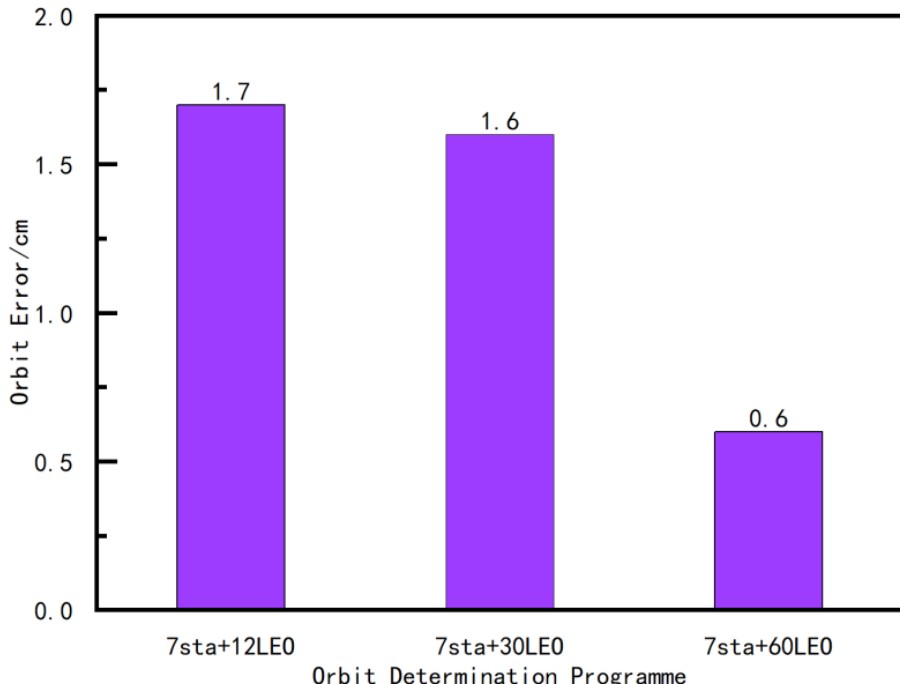

**Figure 8.** 3D accuracy of the LEO POD using ground stations combined with different LEO constellations.

As shown in Figure 9, in the seven sta scheme, the orbit accuracy of the BDS-3 satellites is in the decimeter or meter range. In the seven sta + LEO12 scheme, the orbit error of the BDS-3 satellites has been increased from decimeters to centimeters. This shows that the LEO satellite can indeed improve the orbit accuracy of BDS-3. The centimeter accuracy can be achieved using only seven ground monitoring stations and the LEO12 constellation. The experiments show that the LEO satellite, as a highly dynamic mobile station, participates in the orbit of BDS-3, which causes rapid changes in the tracking geometry between satellites and increases the amount of observation data. However, it is worth noting that, as the number of LEO constellation satellites increases, the orbit accuracy gradually improves, but the accuracy improvement becomes smaller and smaller. This is related to the LEO force model and the accuracy limit of orbit determination.

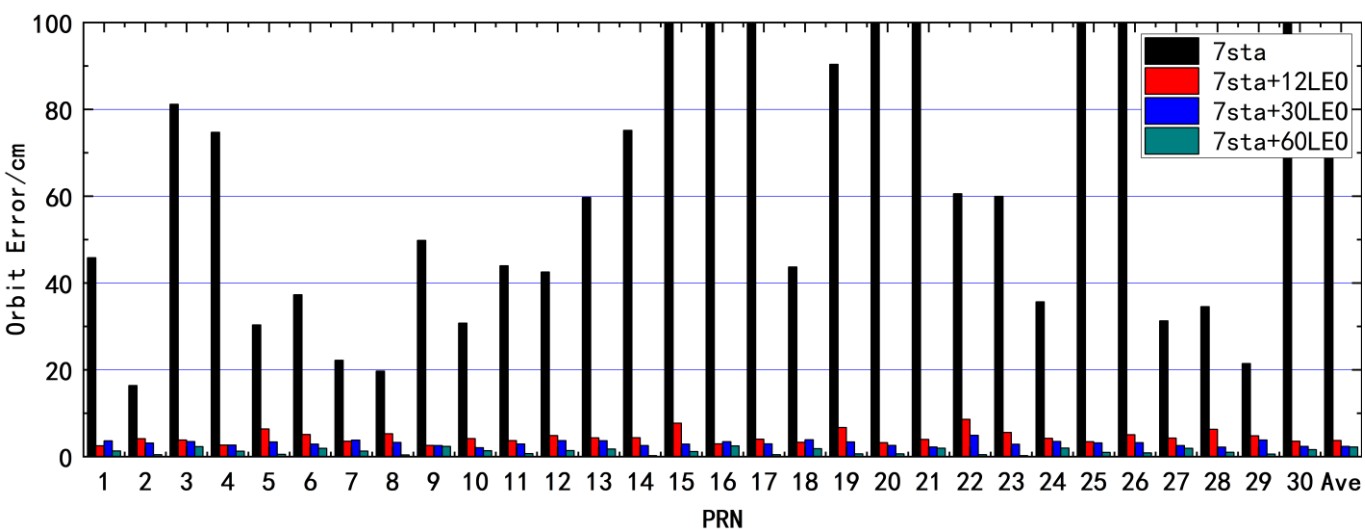

**Figure 9.** The 3D accuracy of the BDS-3 POD using ground stations combined with different LEO constellations.

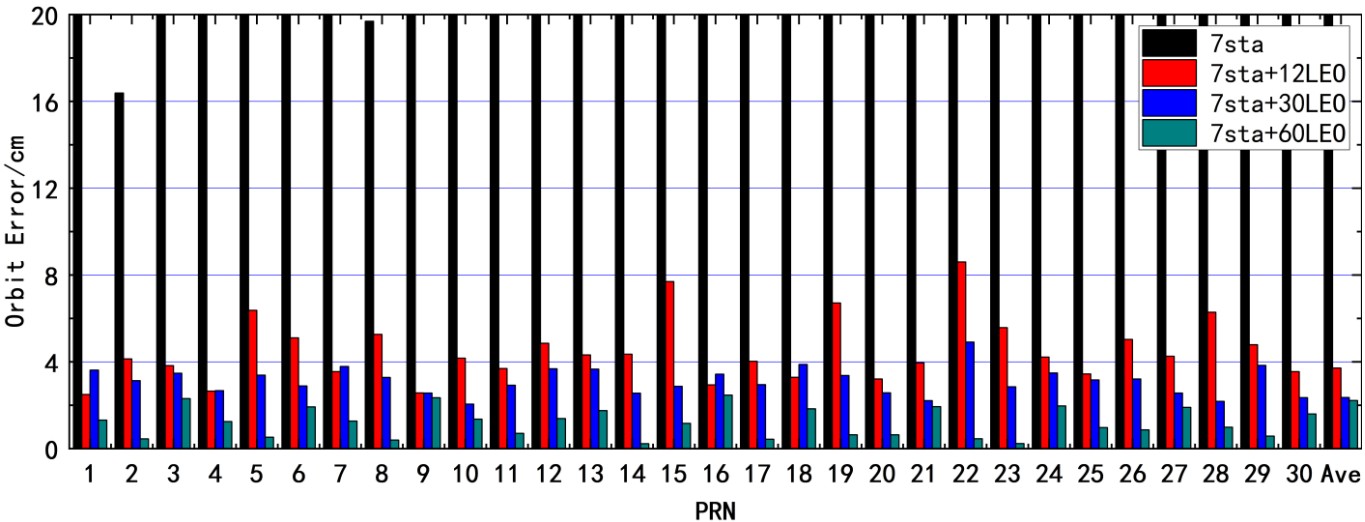

**Figure 10.** The 3D accuracy of the BDS-3 POD using ground stations combined with different LEO constellations (Upper limit 20 cm).

Four different LEO satellite distribution schemes of 7 sta, 7 sta and 12 LEO, 7 sta and 30 LEO, and 7 sta and 60 LEO were used to carry out time synchronization experiments, and as shown in Figure 11, resulting in the average values of RMS1 statistical accuracy of clock offset being 0.72 ns, 0.37 ns, 0.25 ns, and 0.10 ns, respectively. As shown in Figure 12 the average values of RMS2 statistical accuracy of clock offset are 0.68 ns, 0.33 ns, 0.21 ns, and 0.07 ns, respectively. The results show that LEO satellites can improve the accuracy of the BDS-3 satellite's clock offset solution. Compared to the results of clock offset using only ground stations, the addition of 12 LEO satellites resulted in an improvement of 49% for RMS1 and 52% for RMS2, the addition of 30 LEO satellites resulted in an improvement of 66% for RMS1 and 70% for RMS2, and the addition of 60 LEO satellites resulted in an improvement of 87% for RMS1 and 90% for RMS2.

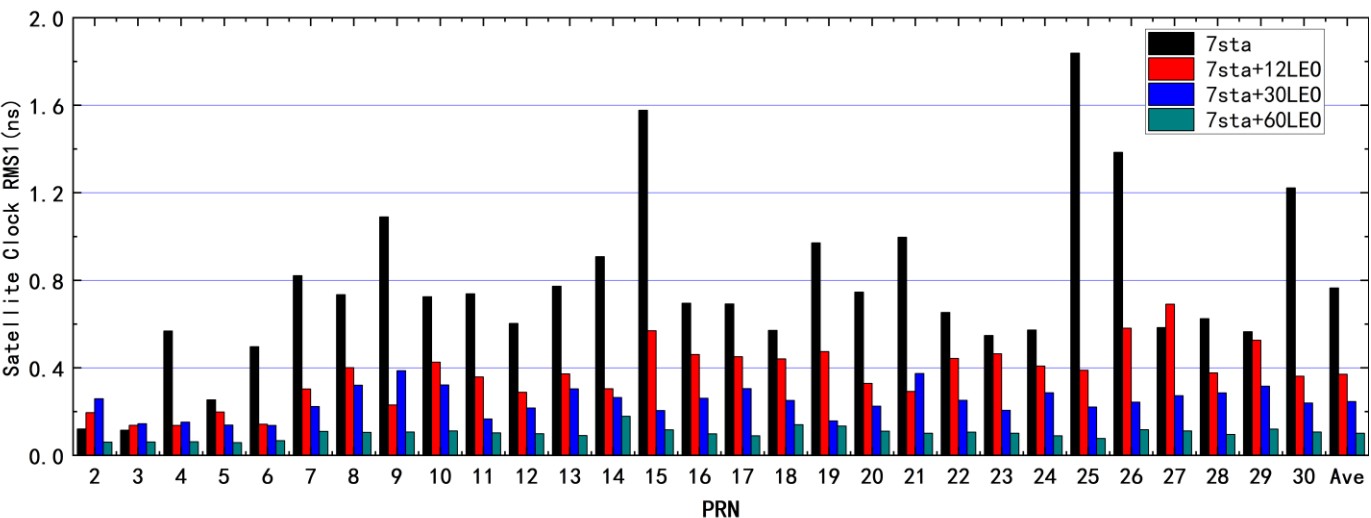

**Figure 11.** RMS1 of the BDS-3 clock offset using ground stations combined with different LEO constellations.

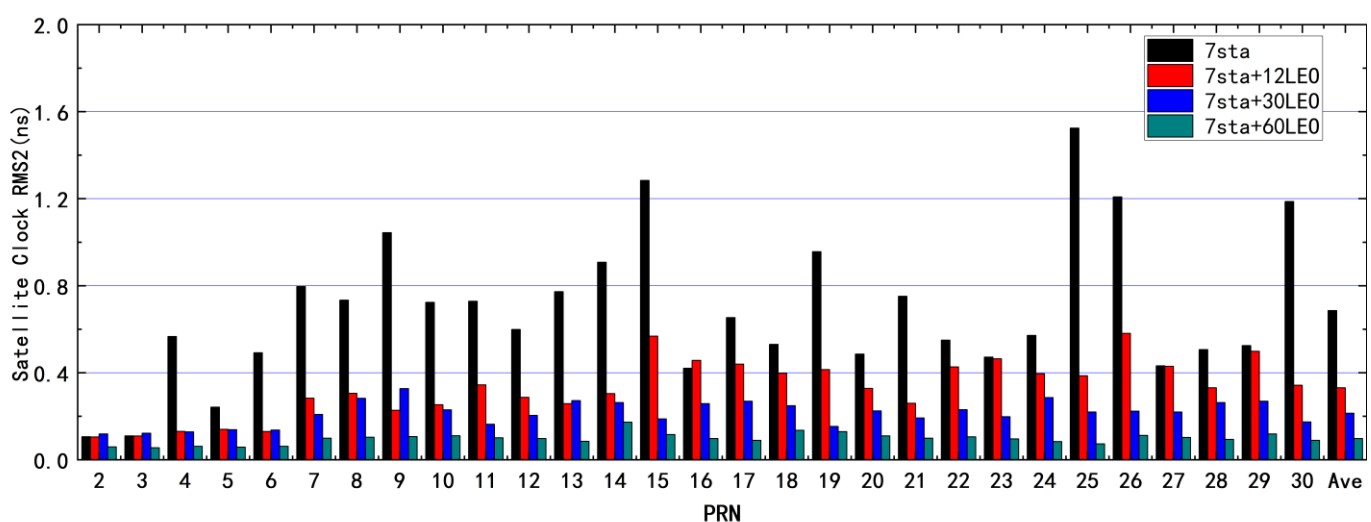

**Figure 12.** RMS2 of the BDS-3 clock offset using ground stations combined with different LEO constellations.

## 4. Discussion

In the above experiment, the LEO constellation and seven ground stations are used in order to integrate orbit determination, and so that the centimeter-level orbital accuracy can be obtained. Only the seven sta scheme tracks accuracy at the decimeter or meter level. The LEO satellite can act as a mobile station, significantly enhancing the diversity of the observation data, while increasing the speed of the geometric observation shape changes and improves the accuracy of BDS-3 POD.

We study the accuracy of BDS-3 orbit determination and time synchronization with simulated observation data of ground stations, LEO satellites, and ISL. According to the simulation experiment results, the LEO constellation enhances the BDS3 precise orbit determination, in the case of a few ground monitoring stations, better than ISL. A LEO constellation can add a large amount of observation information; while LEO satellite flight speed is high, satellite geometry changes faster, and LEO onboard observation is not affected by tropospheric delay error. ISL observation is not affected by atmospheric error, the measurement accuracy is higher, but the corresponding amount of observation is lessened. Both schemes now rely on ground stations to reduce the overall drift of the

constellation. With the rapid development of the LEO constellation, the POD accuracy of integrated LEO satellites can be optimized and improved by greater algorithms. If the ISL data and LEO data are used at the same time, the shortcomings of each could be reduced and the POD accuracy of BDS-3 can be improved.

This study is based on simulated observational data and does not fully reflect the real-world factors that affect measurements, such as the geometry of star-ground tracking, the attitude model of LEO satellites, solar radiation pressure, and antenna calibration. Since the LEO satellites are located at different locations than those of ground-based monitoring stations, they are more susceptible to the effects of the space environment, which can negatively affect the POD. These issues need further research to improve the POD accuracy and time synchronization accuracy of LEO satellites and BDS-3 satellites.

## 5. Conclusions

In the study, simulated BDS-3 ISL data and different LEO constellations were used for integrated POD with ground stations to analyze the accuracies of POD and time synchronization. In the case of a small number of ground monitoring stations, the enhancements of ISL observations and LEO satellite onboard observations for BDS-3 POD are analyzed, respectively.

The results show that the three-dimensional average orbit accuracy is 95.1 cm, 3.1 cm, 2.4 cm, and 2.0 cm for the BDS-3 constellation in the four schemes of 7 sta, 7 sta + 12 LEO, 7 sta + 30 LEO, and 7 sta + 60 LEO, respectively. The clock offset RMS1 is 0.71 ns, 0.37 ns, 0.25 ns, and 0.1 ns, and the RMS2 is 0.6 ns, 0.33 ns, 0.21 ns, and 0.07 ns, respectively. The introduction of LEO satellites has significantly improved the BDS-3 orbit accuracy. With the increase in the number of LEO satellites, BDS-3 orbit accuracy improvement becomes smaller and smaller, while the time required for calculation increases. In the ground station and ISL-integrated orbit determination experiments, using seven ground-monitoring stations and ISL data, the average orbit accuracy of BDS-3 can reach the centimeter level and the time synchronization can reach the nanosecond level. The average orbit accuracy of the seven sta scheme is 74.8 cm, the RMS1 is 0.89 ns, and the RMS2 is 0.83 ns. The average orbit accuracy of the seven sta and ISL scheme is 3.5 cm, the RMS1 is 0.3 ns, and the RMS2 is 0.29 ns. The combination of the ground station and the LEO satellite can greatly improve the time synchronization accuracy and POD accuracy of the BDS-3 satellites. The orbital accuracy of the 7 sta + ISL scheme and the 7 sta + 12 LEO scheme is comparable. This is because of the low orbit of LEO satellites, their fast speed, and the large amount of observation data that can be observed, while the amount of data in the ISL depends on the number of satellites in the BDS-3 constellation.

**Author Contributions:** Conceptualization, X.S. and Z.L.; methodology, X.S. and B.X.; software, B.X. and X.S.; validation, J.C., Q.L., M.S. and Y.X.; formal analysis, Q.L. and Y.X.; investigation, J.C.; resources, X.S. and T.G.; data curation, B.X.; writing—original draft preparation, B.X.; writing—review and editing, B.X. and X.S.; visualization, Z.M.; supervision, Z.L.; funding acquisition, X.S. All authors have read and agreed to the published version of the manuscript.

**Funding:** This work was financially supported by the Ministry of Education-China Mobile Scientific Research Fund Project (MCM2020-J-1) and the Shandong Provincial Natural Science Foundation (Grant number ZR2021QD131).

**Conflicts of Interest:** The authors declare no conflict of interest.

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
