# Peer review of "Analysis on BDS-3 Autonomous Navigation Performance Based on the LEO Constellation and Regional Stations"

_remotesensing, doi:10.3390/rs15123081_

Round 1
Reviewer 1 Report
It is of great value to study the performance of BDS-3 autonomous navigation based on the LEO satellites and regional stations. This paper systematically analyzes theoretically the method of BDS-3 POD and time synchronization based on LEO satellites and regional stations. Verify the accuracy of BDS-3 POD and time synchronization by different types of LEO constellations and ISL data. At the same time, evaluated the performance of BDS-3 autonomous navigation under the use of LEO constellation and ISL. The authors provide reliable experimental conclusions on the autonomous navigation of BDS-3 using LEO constellations or ISL data. The methods and results are quite new and interesting to the autonomous navigation community. The paper was able to meet publication requirements with revisions. I hope that the following comments can help to improve the manuscript.
1. Methodology and equations
(1) In Line 18, ‘(LEO12, LEO30, LEO60, LEO150)’, the LEO 150 constellation is not mentioned in this paper.
(2) In section 3.1, of this paper, BDS-3 autonomous navigation experiments were conducted using ISL data. However, there is no POD processing strategy for ISL. Detailed processing strategies are missing in the manuscript.
2. Experiment and results
(1) In section 3 results, the C01 satellite in Figure 3 lacks the 3D RMS of the 7sta+ISL scheme. The authors must recheck the plotted data and redraw Figure 3.
(2) In section 3.2 lines 325-328. ‘fixing accuracy of the three schemes was 95.1 cm, 3.1 cm, 2.4 cm and 2.0 cm respectively’. Should be ‘four schemes’. ‘fixing accuracy’ is inaccurately expression. This paragraph needs to be described more accurately.
(3) There are repetitions in the article. For example, In Line 289, ‘0.152 m, and 0.152 m’, ‘0.152 m’ is reduplicated.
(4) In Line 310, ‘MEO satellites are distributed all over the world.’ Not a native expression.
There were some typos or grammatical errors in this manuscript, please check through the manuscript and polish the language. For example,
(1) In Line 149, ‘Where, dn denotes the vector dimension corresponding to the relevant perturbing force’, There is a problem with the formatting of this paragraph.
(2) In Lines 169 and 207, ‘carrier phase quantities can be expressed as.’, ‘can be expressed as:’. The use of punctuation in these paragraphs is not standard.
Reviewer 2 Report
See attached comments.

See attached comments.
Reviewer 3 Report
In this manuscript, the authors present simulation results regarding the precise orbit determination (POD) process of the BDS-3 satellites and possible improvements in positioning accuracy and time synchronization. The authors compared the accuracy achieved by performing POD in the following cases:
- 7 ground stations
- 7 ground stations and intersatellite link (ISL) between some BDS-3 satellites
- 7 ground stations and 3 LEO constellations (with 12, 30 and 60 satellites).
The results show that a significant improvement can be achieved using ISL and further improvement is given by the inclusion of data from the LEO satellites. This analysis shows what kind of improvement can be expected in these cases, and therefore can be an interesting work to publish. However, according to this reviewer, the manuscript would benefit from extensive rewriting, as it is very difficult to read at present, particularly in some parts of the text. I have tried to point out a few typos (there were many) and a few sentences that would be good to edit, but it would still be good for the authors to give the manuscript a full check. Also, I have some major doubts about something that I did not fully understand concerning the analysis and the result. I report here all my observations.
Major doubts:
Section 3.1: there are two things I did not understand in this section, or at least I did not find in the text:
- which are the parameters estimated with the POD fit in this case? Only the initial conditions of the 30 satellites or also some model parameter? For example, in table 1 two additional estimated parameters are indicated (inter-frequency bias and inter-system bias).
- Did you consider some systematic error in the dynamical model of the satellites?
Both these points should be clearly expressed in the text, as it is important in order to correctly interpret the results;
Line 348: Here it is stated that the result is related to the LEO force model but in this paper no mention is reported about the LEO force model. Are you considering errors in the dynamical model in these simulations? Can you please add more details on why this behavior is related to the LEO force model? It would be also appropriate to describe the dynamical model used for the simulations;
Line 347-348: the statement reported in these lines is not confirmed by the figures: from Figure 9 it is clear that 7sta+LEO60 case gives a huge improvement with respect to the others; clearly, this statement becomes true for a number of LEO satellites larger than 60, but this should be shown in the results.
Figure 9: why for some satellites the orbit error increases by adding more LEO satellites? (for example for C01, C16 and C18 the LEO30 constellation gives a larger error with respect to the LEO12 constellation)
Figure 10: even here there are some cases in which the claim that the LEO satellites improve the time synchronization is not verified: for example for C01 the error increases with LEO12 and increases more with LEO30; for C03, C04, C21, C27 something similar happens;
line 411-412: Additional cases with more LEO satellites should be added to the simulation to demonstrate this statement.
Typos and suggested corrections:
Line 17: the name of the constellations inside the parenthesis can be replaced with the number of satellites for each constellation to convey a more useful information, for example: “… (with 12, 30, 60 and 150 satellites)”
Line 17: is LEO150 a constellation with 150 satellites? It seems that in the results section only LEO constellation with 12, 30 and 60 satellites are considered; where are the results corresponding to the simulation which accounts for the LEO150 constellation?
Line 18: are simulations ïƒ are simulated
Line 22-23: I suggest to avoid starting the sentence with “And..”
Line 25: RMS 1 and RMS 2 are not defined;
Line 27: the LEO150 case is not reported;
Line 31: “7sta” is not defined;
Line 63-65: here, it is stated that the configuration ISL+stations has been already simulated in other works. Can you briefly describe which are the differences in this paper?
Line 74: IGS is not defined
Line 84: GFZ is not defined
Line 85: Duran et al. ïƒ the reference is not cited or not cited appropriately
Line 88: used the use of satellite-based … ïƒ used satellite-based GPS …
Line 94: PPP is not defined
Line 95: Using ïƒ using
Line 102: in these lines it is stated that there are a few studies about the enhancement of the POD accuracy of BDS-3 by large LEO satellite constellations [23]; can you please briefly describe here which is the additional information provided by this analysis with respect to these previous studies?
Line 130: it seems that there is an incomplete sentence ïƒ “… for a given;”
Line 131: I think it would be appropriate to add more details to the following definition for the benefit of the reader: “p denotes the vector of other parameters to be estimated”; for example: “p is the vector of model parameters to be estimated with the POD filter”.
Line 131-132: I was not able to fully understand this sentence, can you please rephrase it? And also, can you please clarify what do you mean by “double motion” and “regressive accelerations”? considering the first part of the equation, it seems that f0 represents the gravitational force of the Earth (monopole term) and that f1 could indicate other perturbations (spherical harmonic expansion of the Earth’s gravity field, non-gravitational perturbations, third-body perturbations etc..)
Line 143: Equation (5) shall include higher order terms (which can be neglected) as “+O(X-X*) (see for example equation 4.2.5, pag. 161 of “Tapley B, Schutz B and Born G 2004 Statistical Orbit Determination (Amsterdam: Elsevier)”)
Line 167: I am not sure that “Ionosphere-eliminate” is correct. You can make the sentence more concise by saying “Combinations of measurements that eliminate the effect of the first-order term in the ionosphere are the most widely used. In this case, the combined pseudo-range and …”
Line 169: if I am not missing something, it seems that the terms in equation 15 are not defined.
Line 172: The definition of XS* is not really clear, can you please clarify that?
Line 172-173: this part is not really easy to understand, if it is possible, I think it would be good to rephrase for the benefit of the reader.
Line 179: I am not sure that all the terms in Xall have been defined previously in the text.
Line 189: Eq 17 is not introduced
Line 195: the term “treshold” is not really appropriate, please rephrase.
Line 208: I think that the matrix P deserves to be defined (and thus the matrices Psta and Pleo).
Line 209: section 2.2 does not exist
Line 211: what do you mean by “error correction model”?
Line 219: I would suggest to eliminate “due to thin atmosphere…” or alternatively to eliminate the “;”
Line 220-222: this sentence is not really clear, please rephrase it.
Line 223: why are you using DE405 instead of the most recent de440 (doi: 10.3847/1538-3881/abd414)? Also, report a reference for the JPL Ephemerides.
Table 1: the content of line 5, 6, 7 of the table is undefined in both columns.
Section 2.3 in general: which are the estimated model parameters in the POD fit in this analysis?
Line 239: “independent” is repeated
Line 242: “.. days. the 7sta …” ïƒ “.. days. The 7sta …”
Line 242: the 7sta scheme shall be defined before its first appearing in the text to avoid generating confusion.
Line 246-248: please, indicate some references that motivated your choice for the noise selection.
Line 249: If I understood well, the ISL is between GEO and MEO, IGSO and MEO, MEO and MEO, IGSO and IGSO. Is that right?
Line 250: “visible condition” ïƒ did you mean “visibility condition” ?
Line 250: “above” is repeated
Line 251: “… the Earth’s surface, Add …” ïƒ “… the Earth’s surface. Add …”
Line 248-252: please consider rephrasing this part as it is very difficult to understand.
Line 252: it would be good for the benefit of the readers to indicate a reference or a motivation for selecting this noise on the observables.
Line 253: I would suggest: “Considering the constraints of deploying monitoring stations, we simulated seven ground monitoring stations …” ïƒ “We simulated seven ground monitoring stations …”
Line 258 and also in the other sections of the text: Shouldn’t “BDS-3 satellite” be “BDS-3 satellites”? if that is correct, please check also in the rest of the manuscript.
Line 260: I think that “ground-track” is more appropriate than “trajectory”
Line 259-264: As this is a very long sentence, I would suggest to split this part in three sentences, one for the description of the characteristics each type of satellite (GEO, IGSO and MEO).
Line 264-265: I would suggest to change to: “Figure 2 shows the BDS-3 satellites ground tracks”.
Table 2: Third line: PRN has not been defined previously. And what are C01, C02, …, C30? It seems to me that this indicates each satellite but it is not described in the text and also in the table. While one can interpret it later, these things need to be defined somewhere to avoid causing confusion.
Line 274: a space is missing “orbit[8]” ïƒ “orbit [8]”
Line 279: “with an orbit accuracy improvement” ïƒ “with an improvement”
Figure 3: Out of curiosity, why do some satellites show a much larger RMS than others (e.g. C03 for GEO and C07, C15 and C25 for MEO)? does this depend on their geometry with respect to ground stations?
Line 295-298: I would suggest to rephrase this part as it is not easy to understand, if possible, avoiding or minimizing repetitions.
Line 298-299: if it is possible, it would be helpful to add more details here.
Line 301: RMS1 and RMS2 are simply the root mean square and the standard deviation.
Figure 6/7: why do you need to plot both RMS1 and RMS2? Does RMS2 give additional information with respect to RMS1? If not, I would suggest keeping only one figure with a bar plot comparing RMS1 (or RMS2) for the 7sta and 7sta+ISL cases (similar to Figure 3).
Line 308: “the time synchronization accuracy is a little worse” ïƒ “the time synchronization accuracy is in average X times worse than …” or “the RMS error is in average X times larger than …”
Line 317: a space is missing “ISLscheme” ïƒ “ISL scheme”
Line 320: BDS3 is referred to as “BDS-3” in the rest of the paper
Table 3: this table does not seem to be needed: the only thing that is different in the three constellations is the number of satellites, this can be easily reported in the text;
Line 326-328: please rephrase, as this sentence does not appear to be well structured;
Line 327: three ïƒ four
Line 331: “Add 1.000 m of random …” ïƒ “We added 1.000 m of random …”
Line 334: shown in Table 3 ïƒ shown in Figure 8
Line 339: I have some concerns about the following sentence: “As shown in Figure 8, in the 7sta scheme, the orbit accuracy of the BDS-3 satellite is in the decimeter or meter range”:
in figure 8 the result for the 7sta scheme is not reported; also, it is difficult to understand what is meant by “decimeter or meter range” … is it decimeter or meter?
Line 340: orbital accuracy ïƒ orbit error
Figure 10: this is just a close-up of Figure 8; the authors are right in noticing that in Figure 8 the different scales of the errors make difficult to note the bars for the LEO configurations, however this does not seem the best way to make it visible (as in this case two figures are used to represent the same thing); an alternative could be to use a single figure in which the bars represent the improvement with respect to the 7sta case (so in this case you would have three bars instead of 4 and the 7sta case would not reported);
Figure 11: the grid in the figure is missing here;
Line 392-395: I was not able to understand this part of the manuscript; do you mean that solar radiation pressure is not considered at all in the dynamical model of the satellites? And what do you mean by “force model”? Also, please rephrase the sentence, something seems to be missing.
Line 413: Integrated ïƒ integrated
Line 416: the ïƒ The
Please, see the comments in the previous box.
Round 2
Reviewer 1 Report
All m concerns have been resolved. I recommend accepting in current form.
Fine
Reviewer 3 Report
Point 1: Response 1, Comment 2: I did not fully understand the answer to this question, and I apologize if my question was not clear. My doubt was whether this simulation is a covariance analysis or not. Did you perturb the dynamical model used to simulate the observables in comparison to the one used for POD?
Point 2: I now understand that the spacecraft orbits are obtained by modeling the solar radiation pressure (SRP) using the ECOM-5 model for BDS-3 and the macro model for LEO satellites. However, I have a question about line 428 where the authors state that this study lacks important factors, including solar radiation pressure. I thought that SRP is modeled according to the details reported in Table 1 and Table 2.
The quality of the English has definitely improved from the original version.
